# Memory-Efficient LLM Training with Online Subspace Descent

**Kaizhao Liang[†], Bo Liu[†], Lizhang Chen[†], Qiang Liu[†]**
†The University of Texas at Austin
{kaizhaol,bliu,lzchen,lqiang}@utexas.edu

## Abstract

Recently, a wide range of memory-efficient LLM training algorithms have gained substantial popularity. These methods leverage the low-rank structure of gradients to project optimizer states into a subspace using projection matrix found by singular value decomposition (SVD). However, convergence of these algorithms is highly dependent on the update rules of their projection matrix. In this work, we provide the *first* convergence guarantee for arbitrary update rules of projection matrix. This guarantee is generally applicable to optimizers that can be analyzed with Hamiltonian Descent, including most common ones, such as LION, Adam. Inspired by our theoretical understanding, we propose Online Subspace Descent, a new family of subspace descent optimizer without SVD. Instead of updating the projection matrix with eigenvectors, Online Subspace Descent updates the projection matrix with online PCA. Online Subspace Descent is flexible and introduces only minimum overhead to training. We show that for the task of pretraining LLaMA models ranging from 60M to 7B parameters on the C4 dataset, Online Subspace Descent achieves lower perplexity and better downstream tasks performance than state-of-the-art low-rank training methods across different settings and narrows the gap with full-rank baselines.[1]

## 1 Introduction

The continual advancement in training large language models (LLMs) presents a compelling challenge in balancing computational efficiency with model performance. As the scope and complexity of these models grow, so does the necessity for innovative strategies that optimize memory usage without compromising the learning capabilities of the model. Recent approaches in low-rank adaptation strategies, including Stochastic Subspace Descent [13], LoRA [11], ReLoRA [15], Gradient Low-Rank Projection (GaLore) [25] and Sketchy [9] , have paved the way for memory-efficient training by utilizing a periodically updated low-rank projection matrix to manage parameter updates. In particular, GaLore and Sketchy both utilize expensive singular value decomposition to determine the projection matrix, whereas stochastic subspace descent suggests using random matrices as projection matrices and provides convergence analysis on convex functions and objectives. However, to the best of our knowledge, no one has offered any guarantee of convergence for this class of methods on non-convex functions and objectives.

In this work, we provide the first convergence guarantee for arbitrary update rules of the projection matrix. This guarantee is significant because it is broadly applicable to a wide range of optimizers that can be analyzed within the Hamiltonian descent framework [18]. By establishing this convergence guarantee, we demonstrate that our approach is not limited to specific or narrowly defined update rules, but can be extended to include many commonly used optimizers in the field. In particular, this

---

[1]Code is available at https://github.com/kyleliang919/Online-Subspace-Descent.

38th Conference on Neural Information Processing Systems (NeurIPS 2024).

---
**Algorithm 1** Online Subspace Descent
---

1: Required: Optimizer `OptimizerW`, learning rate $\epsilon_t^W$, weight decay $\lambda^W$ for model weights $\boldsymbol{W}_t$; and $\{\texttt{OptimizerP}, \epsilon_t^P, \lambda^P\}$ for the projection matrix $\boldsymbol{P}_t$. Proper initialization.
2: **for** iteration $t$ **do**
3:     Calculate gradient $\boldsymbol{G}_t = \nabla L(\boldsymbol{W}_t)$; Update model weights $\boldsymbol{W}_t$ by

$$(\hat{\boldsymbol{\Delta}}_t, \ \hat{\boldsymbol{S}}_t) = \texttt{OptimizerW}(\boldsymbol{P}_t^\top \boldsymbol{G}_t, \hat{\boldsymbol{S}}_{t-1}), \quad \boldsymbol{W}_{t+1} = \boldsymbol{W}_t + \epsilon_t^W (\boldsymbol{P}_t \hat{\boldsymbol{\Delta}}_{t+1} - \lambda^W \boldsymbol{W}_t)$$

4:     Calculate $\boldsymbol{G}_t^P = \nabla L_{\boldsymbol{G}_t}(\boldsymbol{P}_t)$ for $L_{\boldsymbol{G}_t}(\cdot)$ in Eq (6); Update the projection $\boldsymbol{P}_t$ by

$$(\boldsymbol{\Delta}_t^P, \boldsymbol{S}_t^P) = \texttt{OptimizerP}(\boldsymbol{G}_t^P, \ \boldsymbol{S}_{t-1}^P), \qquad \boldsymbol{P}_{t+1} = \boldsymbol{P}_t + \epsilon_t^P (\boldsymbol{\Delta}_t^P - \lambda^P \boldsymbol{P}_t)$$

5: **end for**
6: Remark: We added weight decay as a common heuristic. We recommend using Adam for both optimizers, and set $\epsilon_t^P = \alpha \epsilon_t^W$ with a constant $\alpha$ (e.g., $\alpha = 5$), and $\lambda^W = \lambda^P$. [2]

---

includes popular algorithms such as LION [4] and Adam [12], which are widely used in various machine learning and optimization tasks. Our results therefore offer a robust theoretical foundation for understanding and analyzing the behavior of these optimizers, ensuring their effectiveness and reliability in diverse applications.

Inspired by our theoretical understanding, we introduce a novel family of memory-efficient optimizers named Online Subspace Descent, which incorporates a dynamically changing projection matrix, replacing the conventional periodic updating approach (SVD) with online PCA. By allowing the projection matrix to evolve in response to the changing gradient landscape, Online Subspace Descent enhances the model's ability to navigate the parameter space more effectively. This dynamic adaptation aligns more closely with the natural progression of learning in deep neural networks, which is characterized by changes in the importance of different characteristics and interactions as training progresses. Through extensive experiments and comparative analysis, we demonstrate that our approach presents lower perplexity in pretraining LLaMA models (ranging from 60M to 1B parameters) on the C4 dataset compared to state-of-the-art low-rank training methods, closing the perplexity gap with full-rank baselines on language model pretraining.

## 2   Optimization Background

The training of deep learning models reduces to an optimization problem

$$\min_{\boldsymbol{W}} L(\boldsymbol{W}),$$

where $\boldsymbol{W}$ is the set of weight matrices of the model. For simplicity, we assume $\boldsymbol{W} \in \mathbb{R}^{n \times m}$ is a single matrix of size $(n, m)$ without loss of generality. For notation, we write $\langle \boldsymbol{A}, \boldsymbol{B} \rangle = \text{tr}(\boldsymbol{A}^\top \boldsymbol{B})$ for inner products of matrices, and $\|A\|^2 = \text{tr}(\boldsymbol{A}^\top \boldsymbol{A})$ the Frobenius norm. We use $A \odot B$ to denote the elementwise product, and $A^{\odot 2} = A \odot A$.

**Example 2.1.** *Update rules of common optimizers:*

*Gradient Descent* :    $\boldsymbol{W}_{t+1} = \boldsymbol{W}_t - \epsilon_t \nabla L(\boldsymbol{W}_t),$

     *Momentum* :    $\boldsymbol{W}_{t+1} = \boldsymbol{W}_t - \epsilon_t \boldsymbol{M}_t, \quad\quad\quad \boldsymbol{M}_t = (1 - \beta)\nabla L(\boldsymbol{W}_t) + \beta \boldsymbol{M}_{t-1},$

     *Lion-$\mathcal{K}$ [4]* :    $\boldsymbol{W}_{t+1} = \boldsymbol{W}_t - \epsilon_t \nabla \mathcal{K}(\boldsymbol{N}_t), \quad \boldsymbol{N}_t = (1 - \beta_1)\nabla L(\boldsymbol{W}_t) + \beta_1 \boldsymbol{M}_t$

                                             $\boldsymbol{M}_t = (1 - \beta_2)\nabla L(\boldsymbol{W}_t) + \beta_2 \boldsymbol{M}_{t-1},$

     *Adam [12]* :    $\boldsymbol{W}_{t+1} = \boldsymbol{W}_t - \epsilon_t \dfrac{\boldsymbol{M}_t}{\sqrt{\boldsymbol{V}_t} + e}, \quad \boldsymbol{M}_t = (1 - \beta_{1t})\nabla L(\boldsymbol{W}_t) + \beta_{1t} \boldsymbol{M}_{t-1},$

                                             $\boldsymbol{V}_t = (1 - \beta_{2t})\nabla L(\boldsymbol{W}_t)^{\odot 2} + \beta_{2t} \boldsymbol{V}_{t-1},$

*where $\epsilon_t$ are step sizes, and $\boldsymbol{M}_t, \boldsymbol{V}_t$ are the first and second order momentum, and $\beta, \beta_1, \beta_2$ are momentum coefficients in $(0, 1)$, with $\beta_{it} = \frac{\beta_i - \beta_i^{t+1}}{1 - \beta_i^{t+1}}$, $i = 1, 2$ for $\beta_1, \beta_2 \in (0, 1)$ in Adam, and $\mathcal{K}$ is any convex function with $\nabla K(\mathbf{0}) = \mathbf{0}$ for Lion-$\mathcal{K}$ [4], and Lion [5] uses $\mathcal{K}(\boldsymbol{X}) = \|\boldsymbol{X}\|_{1,1}$ and $\nabla \mathcal{K}(\boldsymbol{X}) = \text{sign}(\boldsymbol{X})$.*

These optimizers can be unifiedly viewed as updating $\boldsymbol{W}_t$ together with an optimizer state $\boldsymbol{S}_t$:

$$\boldsymbol{W}_{t+1} = \boldsymbol{W}_t + \phi_t(\boldsymbol{S}_t), \qquad\qquad \boldsymbol{S}_t = \psi_t(\boldsymbol{S}_{t-1}, \nabla L(\boldsymbol{W}_t)), \qquad\qquad (1)$$

with some mapping $\phi_t, \psi_t$. We have $\boldsymbol{S}_t = \boldsymbol{M}_t$ for momentum and $\boldsymbol{S}_t = \{\boldsymbol{M}_t, \boldsymbol{V}_t\}$ for Adam. Note that both $\boldsymbol{M}_t, \boldsymbol{V}_t$ are of the same size as the model weights $\boldsymbol{W}_t$, resulting in high memory consumption for large models. This issue is particularly pronounced for Adam, which typically yields the best performance for large language models (LLMs) but incurs the highest memory cost due to the need to maintain both $\boldsymbol{M}_t$ and $\boldsymbol{V}_t$. One key challenge is to retain the high performance of Adam while enhancing its memory efficiency.

**Hamiltonian+Descent**   One powerful approach to studying the dynamic properties of optimizers is to examine their continuous-time ODE forms in the limit of infinitesimal step size. The continuous-time forms provide clearer insights into the asymptotic convergence of the algorithm, abstracting away the choices of step size, discretization, and stochastic errors. The underlying logic is that a "sound" optimizer should be guaranteed to converge to local optima of the loss, at least when using sufficiently small step sizes.

Inspired by [4, 18], we observe that the continuous-time form of many common optimizers yields a *Hamiltian+Descent* structure,

$$\begin{aligned}
\frac{\mathrm{d}}{\mathrm{d}t}\boldsymbol{W}_t &= \partial_{\boldsymbol{S}} H(\boldsymbol{W}_t, \boldsymbol{S}_t) - \Phi(\partial_{\boldsymbol{W}} H(\boldsymbol{W}_t, \boldsymbol{S}_t)) \\
\frac{\mathrm{d}}{\mathrm{d}t}\boldsymbol{S}_t &= -\partial_{\boldsymbol{W}} H(\boldsymbol{W}_t, \boldsymbol{S}_t) - \Psi(\partial_{\boldsymbol{S}} H(\boldsymbol{W}_t, \boldsymbol{S}_t)),
\end{aligned} \qquad (2)$$

where $H(\boldsymbol{W}, \boldsymbol{S})$ is a Hamiltonian (or Lyapunov) function that satisfies

$$\min_{\boldsymbol{S}} H(\boldsymbol{W}, \boldsymbol{S}) = L(\boldsymbol{W}), \quad \forall \boldsymbol{W},$$

so that minimizing $L(\boldsymbol{W})$ reduces to minimizing $H(\boldsymbol{W}, \boldsymbol{S})$; and $\Phi(\cdot), \Psi(\cdot)$ are two monotonic mappings satisfying

$$\|\boldsymbol{X}\|_{\Phi}^2 := \langle \boldsymbol{X}, \Phi(\boldsymbol{X}) \rangle \geq 0, \qquad\qquad \|\boldsymbol{X}\|_{\Psi}^2 := \langle \boldsymbol{X}, \Psi(\boldsymbol{X}) \rangle \geq 0, \qquad\qquad \forall \boldsymbol{X}.$$

With $\Phi(\boldsymbol{X}) = \Psi(\boldsymbol{X}) = 0$, the system in (2) reduces to the standard Hamiltonian system that keeps $H(\boldsymbol{W}_t, \boldsymbol{S}_t) = const$ along the trajectory. When adding the descending components with $\Phi$ and $\Psi$, the system then keeps $H(\boldsymbol{W}, \boldsymbol{S})$ monotonically non-decreasing:

$$\begin{aligned}
\frac{\mathrm{d}}{\mathrm{d}t} H(\boldsymbol{W}_t, \boldsymbol{S}_t) &= \left\langle \partial_{\boldsymbol{W}} H_t, \frac{\mathrm{d}}{\mathrm{d}t}\boldsymbol{W}_t \right\rangle + \left\langle \partial_{\boldsymbol{S}} H_t, \frac{\mathrm{d}}{\mathrm{d}t}\boldsymbol{S}_t \right\rangle \\
&= \langle \partial_{\boldsymbol{W}} H_t, \partial_{\boldsymbol{S}} H_t - \Phi(\partial_{\boldsymbol{W}} H_t) \rangle + \langle \partial_{\boldsymbol{S}} H_t, -\partial_{\boldsymbol{W}} H_t - \Psi(\partial_{\boldsymbol{S}} H_t) \rangle \\
&= -\|\partial_{\boldsymbol{W}} H_t\|_{\Phi}^2 - \|\partial_{\boldsymbol{S}} H_t\|_{\Psi}^2 \leq 0,
\end{aligned} \qquad (3)$$

where we write $\partial_{\boldsymbol{W}} H_t = \partial_{\boldsymbol{W}} H(\boldsymbol{W}_t, \boldsymbol{S}_t)$ and similarly for $\partial_{\boldsymbol{S}} H_t$. The main idea is that the cross terms $\langle \partial_{\boldsymbol{W}} H_t, \partial_{\boldsymbol{S}} H_t \rangle$ are canceled, leaving only the negative terms.

**Example 2.2.** *The momentum method yields following continuous-time form and Hamiltonian:*

$$\frac{\mathrm{d}}{\mathrm{d}t}\boldsymbol{W}_t = -\boldsymbol{M}_t, \qquad \frac{\mathrm{d}}{\mathrm{d}t}\boldsymbol{M}_t = a(\nabla L(\boldsymbol{W}_t) - \boldsymbol{M}_t), \qquad with \quad H(\boldsymbol{W}, \boldsymbol{M}) = L(\boldsymbol{W}) + \frac{\|\boldsymbol{M}\|^2}{2a}.$$

**Example 2.3.** *Adam [12] yields the following continuous-time form and Hamiltonian,*

$$\frac{\mathrm{d}}{\mathrm{d}t}\boldsymbol{W}_t = -\frac{\boldsymbol{M}_t}{\sqrt{\boldsymbol{V}_t} + e}, \qquad \frac{\mathrm{d}}{\mathrm{d}t}\boldsymbol{M}_t = a(\nabla L(\boldsymbol{W}_t) - \boldsymbol{M}_t), \qquad \frac{\mathrm{d}}{\mathrm{d}t}\boldsymbol{V}_t = b(\nabla L(\boldsymbol{W}_t)^{\odot 2} - \boldsymbol{V}_t),$$

$$with \quad H(\boldsymbol{W}, \boldsymbol{M}, \boldsymbol{V}) = L(\boldsymbol{W}) + \frac{1}{2a}\left\langle \frac{\boldsymbol{M}}{\sqrt{\boldsymbol{V}} + e}, \boldsymbol{M} \right\rangle,$$

*for which we can show that $\frac{\mathrm{d}}{\mathrm{d}t} H(\boldsymbol{W}_t, \boldsymbol{M}_t, \boldsymbol{V}_t) \leq 0$ when $a \geq b/4$.*

**Example 2.4.** *The Lion-$\mathcal{K}$ optimizer [5, 4] (without weight decay) can be written into*

$$\frac{\mathrm{d}}{\mathrm{d}t}\boldsymbol{W}_t = \nabla \mathcal{K}((1-b)\boldsymbol{M}_t - b\nabla L(\boldsymbol{W}_t)), \qquad \frac{\mathrm{d}}{\mathrm{d}t}\boldsymbol{M}_t = -a(\nabla L(\boldsymbol{W}_t) + \boldsymbol{M}_t),$$

*where $a \geq 0, b \in [0, 1]$ and $\mathcal{K}(\boldsymbol{X})$ is any convex function that attains the minimum at $\boldsymbol{X} = 0$. One of its Hamiltonians that yields the Hamiltonian+descent structure (Eq (13) in Chen et al. [4]) is*

$$H(\boldsymbol{W}, \boldsymbol{M}) = aL(\boldsymbol{W}) + \frac{1}{1-b}\mathcal{K}((1-b)\boldsymbol{M}).$$

# 3 Memory-Efficient Optimizers via Online Subspace Descent

We introduce the idea of constructing memory efficient optimzers by descending in the subspaces that dynamically changes across iterations as motivated by GaLore [25] and Sketchy [9]. We first derive *static* subspace descent by restricting the whole optimization on a subspace (Section 3.1), and then propose to dynamically change the subspace across iterations as a heuristic to attain the optimization in the full space while only using subspace descent (Section 3.2). In particular, we propose to update the subspaces via continuous online PCA like updates to avoids the need of exact SVD like in GaLore and Sketchy (Section 3.2). Finally, we remark in Section 3.3 the heuristic nature of the derivation of the method and highlight the difficulty in theoretical understanding, which motivates our analysis based on Hamiltonian dynamics in Section 4.

## 3.1 Static Subspace Descent

One popular approach to improving memory efficiency is to confine the optimization to a low-dimensional space. To do this, we impose a low rank structure of $\boldsymbol{W} = \boldsymbol{P}\hat{\boldsymbol{W}}$, where $\boldsymbol{P} \in \mathbb{R}^{n \times k}$ is a projection matrix to be determined later and $\hat{\boldsymbol{W}} \in \mathbb{R}^{k \times m}$ is a dimension-reduced parameter. When $k \ll n$, $\boldsymbol{P}$ and $\hat{\boldsymbol{W}}$ are much smaller in size compared to $\boldsymbol{W}$. Now consider

$$\min_{\hat{\boldsymbol{W}}} L(\boldsymbol{P}\hat{\boldsymbol{W}}).$$

Applying the optimizer from (1) to update $\hat{\boldsymbol{W}}$ along with an optimizer state $\hat{\boldsymbol{S}}$, and mapping the update rule $\hat{\boldsymbol{W}}_{t+1} = \hat{\boldsymbol{W}}_t + \phi_t(\hat{\boldsymbol{S}}_t)$ to that of $\boldsymbol{W} = \boldsymbol{P}\hat{\boldsymbol{W}}_t$, we get

$$\boldsymbol{W}_{t+1} = \boldsymbol{W}_t + \boldsymbol{P}\phi_t(\hat{\boldsymbol{S}}_t), \qquad \hat{\boldsymbol{S}}_t = \psi_t(\hat{\boldsymbol{S}}_{t-1}, \boldsymbol{P}^\top \nabla L(\boldsymbol{W}_t)), \qquad (4)$$

where we used the fact that $\nabla_{\hat{\boldsymbol{W}}} L(\boldsymbol{P}\hat{\boldsymbol{W}}) = \boldsymbol{P}^\top \nabla_{\boldsymbol{W}} L(\boldsymbol{W})$. This yields a more memory-efficient optimizer, as the size of $\hat{\boldsymbol{S}}_t$ is proportional to that of $\hat{W}_t$, much smaller than $\boldsymbol{S}_t$ in (1) when $k \ll n$.

## 3.2 Online Subspace Descent

With a static $\boldsymbol{P}$, regardless of its values, the parameter $\boldsymbol{W}$ is restricted to have a low rank structure. Although low rank assumption is proved to be useful for fine-tuning with LoRA-like methods [11], it is often too limited for pre-training or when the desirable model weights are not inherently low-rank.

To address this problem, Zhao et al. [25] suggested to keep the projected updated in (4), but use different $\boldsymbol{P}$ across the iterations:

$$\boldsymbol{W}_{t+1} = \boldsymbol{W}_t + \boldsymbol{P}_t \phi_t(\hat{\boldsymbol{S}}_t), \quad \hat{\boldsymbol{S}}_t = \psi_t(\hat{\boldsymbol{S}}_{t-1}, \boldsymbol{P}_t^\top \nabla L(\boldsymbol{W}_t)), \quad \boldsymbol{P}_{t+1} = \chi_t(\boldsymbol{P}_t, \boldsymbol{W}_t, \hat{\boldsymbol{S}}_t),$$

where $\chi_t$ is a update rule of $\boldsymbol{P}_t$ that will be determined in the sequel. The intuition is to open up different projection directions at different iterations, so that optimization can be conducted in different subspaces across different iterations. This is similar to the update of coordinate descent, except in a continuous fashion. Note that the update of $\boldsymbol{P}_t$ can be done in parallel with that of $(\boldsymbol{W}_t, \hat{\boldsymbol{S}}_t)$, and incurs no slowdown once it is fast enough to not cause a speed bottleneck.

**Example 3.1.** *Examples of common optimizers equipped with online subspace descent:*

$$\begin{aligned} \textit{Gradient Descent}: \quad & \boldsymbol{W}_{t+1} = \boldsymbol{W}_t - \epsilon_t \boldsymbol{P}_t \boldsymbol{P}_t^\top \boldsymbol{G}_t, \quad && \boldsymbol{G}_t = \nabla L(\boldsymbol{W}_t), \\ \textit{Momentum}: \quad & \boldsymbol{W}_{t+1} = \boldsymbol{W}_t - \epsilon_t \boldsymbol{P}_t \hat{\boldsymbol{M}}_t, \quad && \hat{\boldsymbol{M}}_t = (1-\beta)\boldsymbol{P}_t^\top \boldsymbol{G}_t + \beta \hat{\boldsymbol{M}}_{t-1}, \\ \textit{Lion-}\mathcal{K}: \quad & \boldsymbol{W}_{t+1} = \boldsymbol{W}_t - \epsilon_t \boldsymbol{P}_t \nabla \mathcal{K}(\hat{\boldsymbol{N}}_t), \quad && \hat{\boldsymbol{G}}_t = \boldsymbol{P}_t^\top \boldsymbol{G}_t \\ & \hat{\boldsymbol{N}}_t = (1-\beta_1)\hat{\boldsymbol{G}}_t + \beta_1 \hat{\boldsymbol{M}}_t, \quad && \hat{\boldsymbol{M}}_t = (1-\beta_2)\hat{\boldsymbol{G}}_t + \beta_2 \hat{\boldsymbol{M}}_{t-1}, \\ \textit{Adam}: \quad & \boldsymbol{W}_{t+1} = \boldsymbol{W}_t - \epsilon_t \boldsymbol{P}_t \frac{\hat{\boldsymbol{M}}_t}{\sqrt{\hat{\boldsymbol{V}}_t} + e}, \quad && \hat{\boldsymbol{G}}_t = \boldsymbol{P}_t^\top \boldsymbol{G}_t, \\ & \hat{\boldsymbol{M}}_t = (1-\beta_{1t})\hat{\boldsymbol{G}}_t + \beta_{1t}\hat{\boldsymbol{M}}_{t-1}, \quad && \hat{\boldsymbol{V}}_t = (1-\beta_{2t})\hat{\boldsymbol{G}}_t^{\odot 2} + \beta_{2t}\hat{\boldsymbol{V}}_{t-1}. \end{aligned}$$

How Should $\boldsymbol{P}_t$ be Updated? It is useful to draw intuition from the projected gradient descent rule

$$\boldsymbol{W}_{t+1} = \boldsymbol{W}_t - \epsilon_t \boldsymbol{P}_t \boldsymbol{P}_t^\top \boldsymbol{G}_t, \qquad \boldsymbol{G}_t = \nabla L(\boldsymbol{W}_t), \qquad (5)$$

in which $\boldsymbol{P}_t\boldsymbol{P}_t^\top$ can be viewed as a low rank preconditioning of $\boldsymbol{G}_t$. To make it follow the exact gradient descent, we hope to make $\boldsymbol{P}_t\boldsymbol{P}_t^\top\boldsymbol{G}_t$ approximate $\boldsymbol{G}_t$ as much as possible. In Galore, this is achieved by performing singular value decomposition (SVD) on $\boldsymbol{G}_t$ periodically every $T$ iterations:

$$\boldsymbol{P}_t, \_, \_ = \texttt{torch.linalg.svd}(\boldsymbol{G}_{T\lfloor t/T\rfloor}),$$

where $T\lfloor t/T\rfloor$ is the largest multiple of $T$ less than or equal to $t$. However, numerical SVD incurs a large computational cost for very large models. Also, since $\boldsymbol{P}_t$ is fully determined by $\boldsymbol{G}_{T\lfloor t/T\rfloor}$ calculated from a single mini-batch at the last periodic point, it does not incorporate the gradient information from all data in a timely fashion.

In this work, we propose to update $\boldsymbol{P}_t$ in a continuous online fashion that incorporates the most recent gradient information in a timely fashion, without calling torch.linalg.decompositions routines. We view the update of $\boldsymbol{P}_t$ as conducting an online principal component analysis (PCA) based on the streaming of $\{\boldsymbol{G}_t\}$. In particular, we propose to update $\boldsymbol{P}_t$ at time $t$ by minimizing the following PCA objective:

$$L_{\boldsymbol{G}_t}(\boldsymbol{P}) = \left\| \boldsymbol{P}\boldsymbol{P}^\top\tilde{\boldsymbol{G}}_t - \tilde{\boldsymbol{G}}_t \right\|^2 + \lambda \left\| \boldsymbol{P}^\top\boldsymbol{P} - \boldsymbol{I}_{k\times k} \right\|^2, \quad \tilde{\boldsymbol{G}}_t = \frac{\boldsymbol{G}_t}{\|\boldsymbol{G}_t\|}, \tag{6}$$

where $\|\boldsymbol{A}\| = \mathrm{tr}(\boldsymbol{A}^\top\boldsymbol{A})^{1/2}$ and $\boldsymbol{I}_{k\times k}$ is the $k \times k$ identity matrix; we introduced an auxiliary loss to encourage the columns of $\boldsymbol{P}$ to be orthonormal and normalizes $\boldsymbol{G}_t$ to increase stability.

The key property of $L_{\boldsymbol{G}_t}(\boldsymbol{P})$ in (6) is that all its stable local minimum is a global minimum, and $\boldsymbol{P}$ is a global minimum iff $\boldsymbol{P}\boldsymbol{P}^\top\tilde{\boldsymbol{G}}_t$ forms the optimal rank-$k$ approximation of $\tilde{\boldsymbol{G}}_t$ [e.g., 3]; moreover, we have $\boldsymbol{P}^\top\boldsymbol{P} = I_{k\times k}$ at optima when $\lambda > 0$.

Instead of minimizing $L_{\boldsymbol{G}_t}(\boldsymbol{P})$ exactly, to retain computational efficiency, we propose to update $\boldsymbol{P}_t$ by only performing one step of optimization on $L_{\boldsymbol{G}_t}(\boldsymbol{P})$:

$$\boldsymbol{P}_{t+1} = \texttt{OptimizerP.step}(\boldsymbol{P}_t, \ \nabla_{\boldsymbol{P}}L_{\boldsymbol{G}_t}(\boldsymbol{P}_t)),$$

where `OptimizerP.step` can be a favorite optimizer, such as gradient descent or Adam. Note that when using Adam, we introduce a copy of optimizer state $\boldsymbol{S}_t^P$ for $\boldsymbol{P}_t$. See Algorithm 1. Compared to the exact SVD, each online update of $\boldsymbol{P}_t$ here is fast and can be executed in parallel with the $(\boldsymbol{W}_t, \hat{\boldsymbol{S}}_t)$ updates to avoid slowdown.

## 3.3 Difficulty in Theoretical Understanding

The idea above of projecting an arbitrary optimizer with a dynamically changing $\boldsymbol{P}_t$ is heuristically motivated and lacks an immediate rigorous theoretical justification. The main challenge lies in the complex interaction between the update of $\boldsymbol{U}_t$ and the optimization state $\boldsymbol{S}_t$, which could potentially degrade the convergence and other theoretical properties of the original optimizer. A key question is whether we can develop a theoretical framework to understand how $\boldsymbol{P}_t$ impacts the optimizer's convergence behavior and provide guidance for the design of the update rules of $\boldsymbol{P}_t$.

To gain understanding, it is useful to first exam the simple case of projected gradient descent in (5) which does not have an optimizer state ($\boldsymbol{S}_t = \emptyset$). In this case, since $\boldsymbol{P}_t\boldsymbol{P}_t^\top$ is positive semi-finite, the update $\boldsymbol{P}_t\boldsymbol{P}_t^\top\boldsymbol{G}_t$ is always non-increasing direction of $L(\boldsymbol{W})$ for any $\boldsymbol{P}_t$. The algorithm is essentially a variant of coordinate or subspace descent, where $\boldsymbol{P}_t$ defines the subspace on which one step of gradient descent is conduced at iteration $t$. To ensure that (5) finds a local optimum, we mainly need to ensure that $\boldsymbol{P}_t\boldsymbol{G}_t = 0$ only if $\boldsymbol{G}_t = 0$ to prevent the optimizer from stopping prematurely; this is a mild condition that can be satisfied e.g. when $\boldsymbol{P}_t$ is updated by (online) PCA on $\boldsymbol{G}_t$.

Unfortunately, this coordinate-descent-like interpretation does not apply to more advanced optimizers that track a momentum state $\boldsymbol{S}_t$. This is because $\boldsymbol{S}_t$ accumulates the information from the projected gradient $\boldsymbol{P}_\tau\boldsymbol{G}_\tau$ at all earlier iterations $\tau \leq t$. As $\boldsymbol{P}_t$ changes across time, it is unclear whether the gradient projected to different subspaces $\boldsymbol{P}_\tau$ would be coherent with each other, and useful for future updates that are conducted in different subspaces $\boldsymbol{P}_t$ for $t > \tau$. The difficulty is the inertia effect of $\boldsymbol{S}_t$ that entangles the different subspaces, making the dynamic behavior fundamentally more complicated than naive coordinate descent where the descent in different subspaces is uncoupled. This is what we address in Section 4 via the Hamiltonian descent framework.

# 4 Hamiltonian Descent Meets Subspace Descent: A Lyapunov Analysis

In this section, we show a surprising result that the complication outlined above in Section 3.3 *is not a problem* for optimizers that yields the Hamiltonian+descent structure in (2). Our result is two-fold:

• Section 4.1: When applying Online Subspace Descent on systems in (2), the Hamiltonian+descent structure is preserved once the update rule of $\boldsymbol{P}_t$ has a smooth continuous-time limit. Hence, under very mild conditions, Online Subspace Descent equipped with common optimizers like Adam and Lion automatically yield a Lyapunov function and hence benign continuous-time convergence. Moreover, $\boldsymbol{P}_t$ can, in fact, be generalized to an arbitrary linear operator as shown in Section 4.3.

• Section 4.2: For any smooth $\boldsymbol{P}_t$ update rules that eliminates the degenerate case of $\boldsymbol{P}_t^\top \boldsymbol{G}_t = 0$ while $\boldsymbol{G}_t = 0$ at convergence, the online subspace optimizer guarantees to converge in continuous time to a stationary point of the loss $L(\boldsymbol{W})$. This mild condition is satisfied, for example, when $\boldsymbol{P}_t$ is updated by a typical optimizer on the online PCA objective $L_{\boldsymbol{G}_t}(\boldsymbol{P})$.

## 4.1 Online Subspace Descent Preserves the Hamiltonian+Descent Structure

Applying dynamic projection to Hamiltonian descent in (2), we obtain the following systems:

$$
\begin{aligned}
\frac{\mathrm{d}}{\mathrm{d}t}\boldsymbol{W}_t &= \boldsymbol{P}_t \partial_{\hat{\boldsymbol{S}}} H(\boldsymbol{W}_t, \hat{\boldsymbol{S}}_t) - \Phi(\partial_{\boldsymbol{W}} H(\boldsymbol{W}_t, \hat{\boldsymbol{S}}_t)) \\
\frac{\mathrm{d}}{\mathrm{d}t}\hat{\boldsymbol{S}}_t &= -\boldsymbol{P}_t^\top \partial_{\boldsymbol{W}} H(\boldsymbol{W}_t, \hat{\boldsymbol{S}}_t) - \Psi(\partial_{\hat{\boldsymbol{S}}} H(\boldsymbol{W}_t, \hat{\boldsymbol{S}}_t)) \\
\frac{\mathrm{d}}{\mathrm{d}t}\boldsymbol{P}_t &= \Gamma(\boldsymbol{P}_t, \nabla L(\boldsymbol{W}_t)),
\end{aligned}
\tag{7}
$$

where $\Gamma$ specifies the update rule of $\boldsymbol{P}_t$. Following essentially the same derivation as (3), one can show that $H(\boldsymbol{W}, \boldsymbol{S})$ remains a Lyapunov function of (7), regardless of the choice of $\Gamma$:

$$
\begin{aligned}
\frac{\mathrm{d}}{\mathrm{d}t} H(\boldsymbol{W}_t, \hat{\boldsymbol{S}}_t) &= -\|\partial_{\boldsymbol{W}} H_t\|_\Phi^2 - \|\partial_{\boldsymbol{S}} H_t\|_\Psi^2 + \langle \partial_{\boldsymbol{W}} H_t, \boldsymbol{P}_t \partial_{\hat{\boldsymbol{S}}} H_t \rangle - \langle \partial_{\hat{\boldsymbol{S}}} H_t, \boldsymbol{P}_t^\top \partial_{\boldsymbol{W}} H_t \rangle \\
&= -\|\partial_{\boldsymbol{W}} H_t\|_\Phi^2 - \|\partial_{\boldsymbol{S}} H_t\|_\Psi^2 \le 0,
\end{aligned}
\tag{8}
$$

where the key is to use the *adjoint* property of $\boldsymbol{P}$ and $\boldsymbol{P}^\top$ that $\langle \boldsymbol{P}_t \boldsymbol{X}, \boldsymbol{Y} \rangle = \langle \boldsymbol{X}, \boldsymbol{P}_t^\top \boldsymbol{Y} \rangle$, which cancels the crossing terms, independent of the values of $\boldsymbol{P}_t$. There is no requirement on $\Gamma$ here, besides that the derivative in (8) should exist. As shown in Section 4.3, we can generalize (8) by replacing $\boldsymbol{P}_t$ and $\boldsymbol{P}_t^\top$ with a general linear operator $\mathcal{P}_t$ and its adjoint $\mathcal{P}_t^*$.

Please refer to Appendix A for continuous-time Momentum, Lion-$\mathcal{K}$ and Adam with subspace descent and their Hamiltonian functions.

## 4.2 Convergence to Local Optima

In addition to the Lyapunov structure, we need an additional mild condition on the update rule of $\boldsymbol{P}_t$ to ensure the system converges to the local optimum of the loss $L(\boldsymbol{W})$. The main idea is to prevent the system from stopping prematurely before reaching zero gradient $\boldsymbol{G}_t = 0$ by excluding the degenerate case of $\boldsymbol{P}_t \boldsymbol{G}_t = 0$ while $\boldsymbol{G}_t \ne 0$ in the invariant set of the system.

**Assumption 4.1.** *Assume the functions in system (7) are continuously differentiable and*

*i) $\frac{\mathrm{d}}{\mathrm{d}t} H(\boldsymbol{W}_t, \hat{\boldsymbol{S}}_t) = 0$ implies $\hat{\boldsymbol{G}}_t = \boldsymbol{P}_t^\top \nabla L(\boldsymbol{W}_t) = 0$ and $\frac{\mathrm{d}}{\mathrm{d}t} \boldsymbol{W}_t = 0$.*

*ii) When $\boldsymbol{G}_t \equiv \boldsymbol{G} \ne 0$, the set $\{\boldsymbol{P} \colon \boldsymbol{P}^\top \boldsymbol{G} = 0\}$ is not a positive invariant set of $\frac{\mathrm{d}}{\mathrm{d}t} \boldsymbol{P}_t = \Gamma(\boldsymbol{P}_t, \boldsymbol{G}_t)$.*

This is a mild condition. Assumption i) says that the optimizer should stop at a point with $\hat{\boldsymbol{G}}_t = 0$, which is easy to verify for the common optimizers like momentum, Adam, Lion-$\mathcal{K}$. Assumption ii) ensures $\hat{\boldsymbol{G}}_t = 0$ would imply $\boldsymbol{G}_t = 0$ in invariance sets, which is satisfied when for example, $\boldsymbol{P}_t$ is updated by a reasonable optimizer of the online PCA loss that converges to a stable local minimum.

**Theorem 4.2.** *Assume Assumption 4.1 holds. Let $(\boldsymbol{W}_t, \boldsymbol{S}_t, \boldsymbol{P}_t)_t$ be a bounded solution of (7), then all the accumulation points $\{\boldsymbol{W}_t\}$ as $t \to +\infty$ are stationary points of $L(\boldsymbol{W})$.*

*Proof.* By LaSalle's invariance principle, the positive limit set of $(\boldsymbol{W}_t, \boldsymbol{S}_t, \boldsymbol{P}_t)_t$ must be contained in $\mathcal{I}$, where $\mathcal{I} = \{$the union of complete trajectories satisfying $\frac{\mathrm{d}}{\mathrm{d}t} H(\boldsymbol{W}_t, \hat{\boldsymbol{S}}_t) = 0, \forall t \}$.

From the Assumption i), the trajectories contained in $\mathcal{I}$ must satisfy $\frac{d}{dt}\boldsymbol{W}_t = 0$, which implies $\frac{d}{dt}\boldsymbol{G}_t = \frac{d}{dt}\nabla L(\boldsymbol{W}_t) = 0$ and $\hat{\boldsymbol{G}}_t = 0$ and hence $\boldsymbol{G}_t \equiv \boldsymbol{G}$ is a constant with $\boldsymbol{P}_t^\top \boldsymbol{G} = 0$. Moreover, from Assumption ii), we must have $\nabla L(\boldsymbol{W}_t) = \boldsymbol{G}_t \equiv 0$, since otherwise the trajectory is not invariant. As a result, all trajectories in the limit set $\mathcal{I}$ must have $\nabla L(\boldsymbol{W}_t) = 0$. Because $\frac{d}{dt}W_t = 0$, these trajectories are static points of $\boldsymbol{W}_t$. $\qquad\square$

### 4.3 Online Subspace Descent with General Linear Projection Operators

We can generalize the online subspace descent with general linear operators:

$$\frac{d}{dt}\boldsymbol{W}_t = \mathcal{P}_t(\partial_{\hat{\boldsymbol{S}}}H(\boldsymbol{W}_t, \hat{\boldsymbol{S}}_t)) - \Phi(\partial_{\boldsymbol{W}}H(\boldsymbol{W}_t, \hat{\boldsymbol{S}}_t))$$

$$\frac{d}{dt}\hat{\boldsymbol{S}}_t = -\mathcal{P}_t^*(\partial_{\boldsymbol{W}}H(\boldsymbol{W}_t, \hat{\boldsymbol{S}}_t)) - \Psi(\partial_{\hat{\boldsymbol{S}}}H(\boldsymbol{W}_t, \hat{\boldsymbol{S}}_t))$$

$$\frac{d}{dt}\mathcal{P}_t = \Gamma(\mathcal{P}_t, \nabla L(\boldsymbol{W}_t)),$$

where we generalize $\boldsymbol{P}_t$ to be any linear operator $\mathcal{P}_t$ with an adjoint operator $\mathcal{P}_t^*$, satisfying

$$\langle \boldsymbol{X}, \mathcal{P}_t(\boldsymbol{Y}) \rangle = \langle \mathcal{P}_t^*(\boldsymbol{X}), \boldsymbol{Y} \rangle, \quad \forall \boldsymbol{X}, \boldsymbol{Y}.$$

The derivation of Lyapunov follows a similar way:

$$\frac{d}{dt}H(\boldsymbol{W}_t, \hat{\boldsymbol{S}}_t) = -\|\partial_{\boldsymbol{W}}H_t\|_\Phi^2 - \|\partial_{\boldsymbol{S}}H_t\|_\Psi^2 + \langle \partial_{\boldsymbol{W}}H_t, \mathcal{P}_t(\partial_{\hat{\boldsymbol{S}}}H_t) \rangle - \langle \partial_{\hat{\boldsymbol{S}}}H_t, \mathcal{P}_t^*(\partial_{\boldsymbol{W}}H_t) \rangle$$

$$= -\|\partial_{\boldsymbol{W}}H_t\|_\Phi^2 - \|\partial_{\boldsymbol{S}}H_t\|_\Psi^2 \le 0,$$

where the crossing terms are again canceled due to the adjoint property.

As an example of the general framework, consider $\mathcal{P}_t(\boldsymbol{X}) = \boldsymbol{P}_t\boldsymbol{X}\boldsymbol{Q}_t$, where $\boldsymbol{Q}_t$ is another projection matrix applied on the different dimension of $\boldsymbol{X}$ (see also [25]). The adjoint operator of $\mathcal{P}_t$ is $\mathcal{P}_t^*(\boldsymbol{X}) = \boldsymbol{P}_t^\top\boldsymbol{X}\boldsymbol{Q}_t^\top$. This can be verified by

$$\langle \boldsymbol{P}_t\boldsymbol{X}\boldsymbol{Q}_t, \boldsymbol{Y} \rangle = \text{tr}(\boldsymbol{P}_t\boldsymbol{X}\boldsymbol{Q}_t\boldsymbol{Y}^\top) = \text{tr}(\boldsymbol{X}\boldsymbol{Q}_t\boldsymbol{Y}^\top\boldsymbol{P}_t) = \text{tr}(\boldsymbol{X}(\boldsymbol{P}_t^\top\boldsymbol{Y}\boldsymbol{Q}_t^\top)^\top) = \langle \boldsymbol{X}, \boldsymbol{P}_t^\top\boldsymbol{Y}\boldsymbol{Q}_t^\top \rangle.$$

The subspace descent system of this operator is

$$\frac{d}{dt}\boldsymbol{W}_t = \boldsymbol{P}_t\partial_{\hat{\boldsymbol{S}}}H(\boldsymbol{W}_t, \hat{\boldsymbol{S}}_t)\boldsymbol{Q}_t - \Phi(\partial_{\boldsymbol{W}}H(\boldsymbol{W}_t, \hat{\boldsymbol{S}}_t))$$

$$\frac{d}{dt}\hat{\boldsymbol{S}}_t = -\boldsymbol{P}_t^\top\partial_{\boldsymbol{W}}H(\boldsymbol{W}_t, \hat{\boldsymbol{S}}_t))\boldsymbol{Q}_t^\top - \Psi(\partial_{\hat{\boldsymbol{S}}}H(\boldsymbol{W}_t, \hat{\boldsymbol{S}}_t))$$

$$\frac{d}{dt}\boldsymbol{P}_t = \Gamma_P(\boldsymbol{P}_t, \boldsymbol{Q}_t, \nabla L(\boldsymbol{W}_t))$$

$$\frac{d}{dt}\boldsymbol{Q}_t = \Gamma_Q(\boldsymbol{P}_t, \boldsymbol{Q}_t, \nabla L(\boldsymbol{W}_t)),$$

where $\boldsymbol{P}_t, \boldsymbol{Q}_t$ can be updated jointly via an online SVD on $\boldsymbol{G}_t$.

Another linear operator that involves two matrices is $\mathcal{P}_t(\boldsymbol{X}) = \boldsymbol{P}_t\boldsymbol{X} + \boldsymbol{X}\boldsymbol{Q}_t$, which yields $\mathcal{P}_t^*(\boldsymbol{X}) = \boldsymbol{P}_t^\top\boldsymbol{X} + \boldsymbol{X}\boldsymbol{Q}_t^\top$.

## 5 Experiment

We answer a number of key questions with pretraining experiments of LLaMA [22] on the C4 dataset [20]. All experiments except for large 7B experiments are conducted on a *single* NVIDIA A100 GPU.

### 5.1 Why do we Need Online Subspace Descent?

Overall, Online Subspace Descent offers two major advantages over previous methods that rely on SVD, better convergence and lower overhead. In this section, we discuss both in detail.

First, Online Subspace Descent closes the gap between the state-of-the-art low-rank method and full rank baseline uniformly across different model sizes, as shown in figure 1. A highlight amongst these results is LLaMA 1B (SS 256). As shown in table 1, Online Subspace Descent attains significant improvement over GaLore in perplexity, while consuming a similar amount of GPU memory (8.64 GB v.s 9.01 GB). One additional observation in 1 shows as model size and sequence length grow, Online Subspace Descent becomes more effective. We hypothesize that this is due to the higher intrinsic rank of the underlying optimization problem in larger models. Hence, the positive impact on the convergence of the online update of $P_t$ becomes more obvious. See more details in Appendix B.

| Method | Perplexity($\downarrow$) | | |
|---|---|---|---|
| | **60M** | **350M** | **1B** |
| 8bit-AdamW (Full Rank) | 32.75 | 30.43 | 29.40 |
| GaLore (Rank = 512) | 57.03 | 44.34 | 35.52 |
| **Ours** (Rank = 512) | **56.12** | **43.67** | **31.30** |

Table 1: Pretraining LLaMA 1B with a sequence length of 256 and for 10K steps, perplexity was reported as the training average of the last 10 steps. AdamW8bit serves as the base optimizer for both.

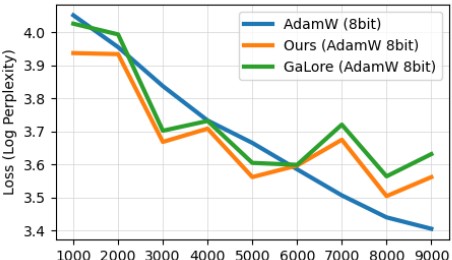

Figure 1: Validation perplexity of LLaMA 1B with sequence length 256, rank 512 for 10K steps.

Another favorable characteristic of Online Subspace Descent is its minimum overhead. In figure 2, we measure and analyze the execution time of SVD and online PCA on a popular data center GPU (A100) and a consumer GPU (RTX 3090). The typical Pytorch implementation of SVD can be up to 142 times slower than running a single-step online PCA on representative weight tensors from LLaMA architectures. Online PCA is fast because it is implemented as a single optimization step with respect to a simple loss function. Hence, each step of online PCA can be cleverly scheduled and hidden in the weight optimization step when executed in parallel, whereas SVD is too expensive to be hidden.

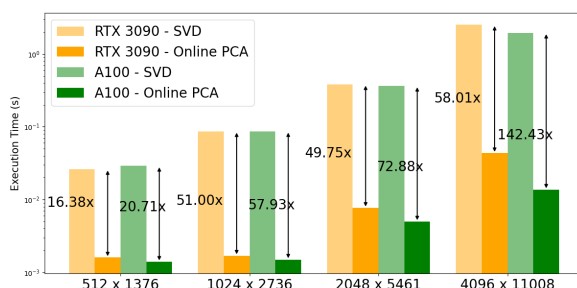

Figure 2: The execution time of torch.svd and that a single-step backward() call for online PCA in PyTorch, on matrices of typical shapes in linear layers in the LLaMA 60M to 7B. Thanks to the high speed of single-step online PCA, $P_t$ updates can be executed in parallel with weight updates, adding no overhead to the training process. In contrast, SVD incurs significant overhead as the model and weight tensor sizes increase.

### 5.2 What Rank Should we Pick for *Online Subspace Descent*?

We conduct an ablation study on the rank of Online Subspace Descent. Figure 3 shows that the final perplexity is inversely correlated with rank: higher ranks result in lower convergent perplexity. However, the rate of reduction of perplexity decreases as the rank increases, eventually reaching a saturation point. We propose an intuitive explanation for this phenomenon. In language modeling, high-frequency tokens can be effectively learned with low-rank training. However, learning lower-frequency tokens requires higher ranks. Once these lower-frequency tokens are adequately learned, further increasing the rank does not significantly decrease perplexity. In conclusion, given sufficient time and resources, higher ranks yield better performance for Online Subspace Descent. It is recommended that the highest rank be selected until the perplexity reduction saturates.

### 5.3 What are the Best Hyperparameters?

$\alpha$ **and** $\lambda$: The parameter $\alpha$ controls the update speed of $P_t$, while $\lambda$ determines the regularization strength on the optimization objective of $P_t$. Empirically, we find that the result is not sensitive to $\lambda$

for small models (60M). and set $\lambda = 0.1$ for all subsequent experiments. We find that $\alpha$ must be kept small to avoid instability (Figure 3), and we set $\alpha = 5$ for all experiments.

**Learning rate**: For the small model (60M), learning rate choices are more flexible, producing similar results. However, for larger models (350M, 1B), we recommend using a learning rate that is 10 times smaller, specifically 0.001. Larger learning rates cause unrecoverable spikes and instability, a general characteristic observed across all methods. See additional hyperparameter choices in Appendix B.

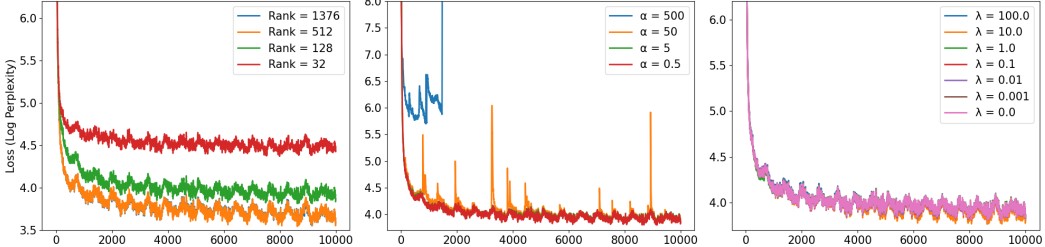

Figure 3: From left to right are loss curves of 10K steps on LLaMA 60M: leftmost is the sweep of rank, middle is the sweep of $\alpha$ and rightmost is the sweep of $\lambda$.

## 5.4 Can *Online Subspace Descent* be Applied to Different Optimizers?

One straightforward extension of Online Subspace Descent is to apply it to other base optimizers beyond AdamW8bit. We conduct ablation studies on LION [6] and Adafactor [21], finding that Online Subspace Descent behaves similarly to how it does with AdamW8bit. Despite the initial observation that updating $\boldsymbol{P}_t$ with AdamW8bit consistently yields better results, we discover that updating $P_t$ with simple SGD can achieve similar performance.

| Method | GaLore | | Ours | | | |
|---|---|---|---|---|---|---|
| | Lion | Adaf. | Lion+Lion | Adaf.+Adaf. | Lion+AdamW | Adaf.+AdamW |
| **Perplexity** | 46.90 | 34.32 | 57.97 | 47.61 | **44.76** | **34.15** |

Table 2: LLaMA 60M on C4 with sequence length 1024, with optimizers on $\boldsymbol{P}_t$ and $\boldsymbol{W}_t$, denote as "Ours {$\boldsymbol{W}_t$ optimizer} + {$\boldsymbol{P}_t$ optimizer}". Adaf., and Adam refer to Adafactor and 8bit-AdamW, respectively.

## 5.5 Can Online Subspace Descent Scale to Larger Model?

We pretrain from scratch a 7B LLaMA model on the C4 dataset for 10K steps, where the $\boldsymbol{P}_t$ matrix is updated by SGD. The perplexity is the lower the better. The final perplexity and training wall-clock time are provided in Table 3. We further provide the downstream evaluation of the pretrained checkpoints using Galore and our method on the GLUE benchmark in Table 4. Our method consistently outperforms Galore when the model size scales up.

| Method | Perplexity | Wall Clock Time (hours) |
|---|---|---|
| Galore | 51.21 | 9.7439 |
| Ours | **43.72** | **7.1428** |

Table 3: Perplexity and Wall Clock Time for 7B models pretrained on C4 for 10K steps. Lower perplexity is better. Online Subspace Descent can be upto 1.3x faster than GaLore.

# 6 Related Works

We discuss related works on memory-efficient optimization and low-rank adaptation techniques.

| Method | MRPC | RTE | SST2 | MNLI | QNLI | QQP | AVG |
|--------|------|-----|------|------|------|-----|-----|
| Galore | 0.6838 | **0.5018** | 0.5183 | 0.3506 | 0.4946 | 0.3682 | 0.4862 |
| Ours | **0.6982** | 0.4901 | **0.5233** | **0.3654** | **0.5142** | **0.3795** | **0.4951** |

Table 4: Standardized GLUE evaluation for 7B model checkpoints using eval-harness. Results are reported for various downstream tasks.

**Low-Rank Adaptation**    Low-Rank Adaptation (LoRA) [11] adds a low-rank adaptor to speficic linear layers in a model, and finetune only the low-rank adaptor. As the adaptors are small, LoRA is widely applied for finetuning large models. Many variants have been proposed since LoRA, including support for multi-task learning  Wang et al. [23] and further memory reductions Dettmers et al. [8]. Notably, Lialin et al. [15] proposed ReLoRA for pretraining, requiring a full-rank training warmup to match standard performance levels. It's important to note that LoRA is fundamentally distinct from subspace descent. While subspace descent optimizes within the original model parameter space, LoRA focuses its optimization efforts within the space of the adaptors.

**Memory-Efficient Optimization**    Several approaches aim to reduce memory costs associated with gradient statistics in adaptive optimization algorithms [21, 2, 7]. In particular, Adafactor [21] factorizes the second-order statistics by a row-column outer product and update the factorized bases on the fly, hence achieving a sub-linear memory cost. K-Fac [19] presents a factorized approximation of the Fisher information matrix which leads to a sublinear natural gradient method. More recently, Feinberg et al. [9] observes that the spectra of the Kronecker-factored gradient covariance matrix in deep learning (DL) training tasks are concentrated on a small leading eigenspace and propose to maintain a matrix preconditioner using the frequent directions sketch. However, their method requires conducting the eigendecomposition at every step, which can be costly for large models. Other than factorization methods, quantization techniques [7, 1, 24, 16] are also widely used, where the gradient (or the momentum and the preconditioner) are directly quantized to tradeoff performance for memory. Fused gradient computation method [17] have also been used to minimize memory costs during training. GaLore [25] is the most relevant work to ours. GaLore focuses on low-rank gradient structures, reducing memory costs for both first and second-order statistics. Our method can be viewed as a general extension to GaLore where we replace the infrequent SVD by a continuous subspace descent [14, 10]. As a result, our method not only provides a more general framework to study memory-efficient subspace descent, but is also more performant than GaLore in practice.

# 7    Conclusion

In conclusion, we provide the first convergence guarantee for arbitrary update rules of projection matrix, applicable to a range of optimizers that can be analyzed using Hamiltonian Descent, including common ones like LION, AdamW, and Adafactor. Inspired by this theoretical foundation, we introduce Dynamic Subspace Descent, a novel family of subspace descent optimizers that eschews SVD in favor of online PCA for updating projection matrix. Dynamic Subspace Descent is both flexible and minimally intrusive, and our experiments show that it achieves lower perplexity in pretraining LLaMA models (ranging from 60M to 1B parameters) on the C4 dataset compared to state-of-the-art low-rank training methods, while also closing the perplexity gap with full-rank baselines.

For future research, we propose several open and intriguing questions: (1) Are there alternative methods for updating projection matrix that could accelerate convergence? (2) What is the impact of weight decay on convergence in Dynamic Subspace Descent? (3) Can low-rank gradients and updates be combined with dynamic low-rank weights (e.g., Mixture of Experts) to further enhance training efficiency? (4) Can this method be applied to problems beyond language modeling? We hope that our work provides a strong foundation for exploring these questions.

# 8 Acknowledgment

The research is conducted in Statistics & AI group at UT Austin, which receives supports in part from NSF CAREER1846421, SenSE2037267, Office of Navy Research, and NSF AI Institute for Foundations of Machine Learning (IFML).

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

# A    Hamiltonian Examples

**Example A.1.** *Momentum + Online Subspace Descent is*

$$\frac{\mathrm{d}}{\mathrm{d}t}\boldsymbol{W}_t = -\boldsymbol{P}_t\hat{\boldsymbol{M}}_t, \qquad \hat{\boldsymbol{G}}_t = \boldsymbol{P}_t^\top\nabla L(\boldsymbol{W}_t), \qquad \frac{\mathrm{d}}{\mathrm{d}t}\hat{\boldsymbol{M}}_t = a(\hat{\boldsymbol{G}}_t - \hat{\boldsymbol{M}}_t),$$

*with Lyapunov function* $\quad H(\boldsymbol{W}, \hat{\boldsymbol{M}}) = L(\boldsymbol{W}) + \dfrac{\|\hat{\boldsymbol{M}}\|^2}{2a}, \;$ *for which*

$$\frac{\mathrm{d}}{\mathrm{d}t}H(\boldsymbol{W}_t, \hat{\boldsymbol{M}}_t) = -\nabla L(\boldsymbol{W}_t)^\top\boldsymbol{P}_t\hat{\boldsymbol{M}}_t + \hat{\boldsymbol{M}}_t^\top(\boldsymbol{P}_t^\top\nabla L(\boldsymbol{W}_t) - \hat{\boldsymbol{M}}_t) = -\left\|\hat{\boldsymbol{M}}_t\right\|_2^2 \le 0.$$

**Example A.2.** *Adam + Online Subspace Descent is*

$$\frac{\mathrm{d}}{\mathrm{d}t}\boldsymbol{W}_t = \boldsymbol{P}_t\frac{\hat{\boldsymbol{M}}_t}{\sqrt{\hat{\boldsymbol{V}}_t}+e}, \quad \hat{\boldsymbol{G}}_t = \boldsymbol{P}_t^\top\nabla L(\boldsymbol{W}_t), \quad \frac{\mathrm{d}}{\mathrm{d}t}\hat{\boldsymbol{M}}_t = a(\hat{\boldsymbol{G}}_t - \hat{\boldsymbol{M}}_t), \quad \frac{\mathrm{d}}{\mathrm{d}t}\hat{\boldsymbol{V}}_t = b(\hat{\boldsymbol{G}}_t^2 - \hat{\boldsymbol{V}}_t).$$

*with Lyapunov function* $\quad H(\boldsymbol{W}, \boldsymbol{M}, \boldsymbol{V}) = L(\boldsymbol{W}) + \dfrac{1}{2a}\left\langle \dfrac{\hat{\boldsymbol{M}}}{\sqrt{\hat{\boldsymbol{V}}}+e}, \; \hat{\boldsymbol{M}} \right\rangle, \;$ *for which*

$$\frac{\mathrm{d}}{\mathrm{d}t}H(\boldsymbol{W}_t, \hat{\boldsymbol{M}}_t, \hat{\boldsymbol{V}}_t)$$

$$= -\left\langle \boldsymbol{G}_t, \boldsymbol{P}_t\frac{\hat{\boldsymbol{M}}_t}{\sqrt{\hat{\boldsymbol{V}}_t}+e} \right\rangle + \frac{1}{a}\left\langle \frac{\hat{\boldsymbol{M}}_t}{\sqrt{\hat{\boldsymbol{V}}_t}+e}, \; a(\boldsymbol{P}_t^\top\boldsymbol{G}_t - \hat{\boldsymbol{M}}_t) \right\rangle - \frac{b}{4a}\left\langle \frac{\hat{\boldsymbol{M}}_t^{\odot 2}}{\sqrt{\hat{\boldsymbol{V}}_t}\odot(\sqrt{\hat{\boldsymbol{V}}_t}+e)^{\odot 2}}, \; (\hat{\boldsymbol{G}}_t^{\odot 2} - \hat{\boldsymbol{V}}_t) \right\rangle$$

$$= -\left\langle 1 - \frac{b}{4a}\frac{\sqrt{\hat{\boldsymbol{V}}_t}}{\sqrt{\hat{\boldsymbol{V}}_t}+e}, \; \frac{\hat{\boldsymbol{M}}_t^{\odot 2}}{\sqrt{\hat{\boldsymbol{V}}_t}+e} \right\rangle - \frac{b}{4a}\left\langle \frac{\hat{\boldsymbol{M}}_t^{\odot 2}}{\sqrt{\hat{\boldsymbol{V}}_t}\odot(\sqrt{\hat{\boldsymbol{V}}_t}+e)^{\odot 2}}, \; \hat{\boldsymbol{G}}_t^{\odot 2} \right\rangle$$

$$\le -\left(1 - \frac{b}{4a}\right)\left\|\frac{\hat{\boldsymbol{M}}_t}{\sqrt{\sqrt{\hat{\boldsymbol{V}}_t}+e}}\right\|^2 - \frac{b}{4a}\left\|\frac{\hat{\boldsymbol{M}}_t\hat{\boldsymbol{G}}_t}{\sqrt[4]{\hat{\boldsymbol{V}}_t}(\sqrt{\hat{\boldsymbol{V}}_t}+e)}\right\|^2 \le 0,$$

*where we assume* $a \ge b/4$.

**Example A.3.** *The Lion-$\mathcal{K}$ + Online Subspace Descent is*

$$\frac{\mathrm{d}}{\mathrm{d}t}\boldsymbol{W}_t = \boldsymbol{P}_t\nabla\mathcal{K}((1-b)\hat{\boldsymbol{M}}_t - b\hat{\boldsymbol{G}}_t), \quad \frac{\mathrm{d}}{\mathrm{d}t}\boldsymbol{M}_t = -a(\hat{\boldsymbol{G}}_t + \hat{\boldsymbol{M}}_t), \quad \hat{\boldsymbol{G}}_t = \boldsymbol{P}_t^\top\nabla L(\boldsymbol{W}_t)$$

*Consider the Hamiltonian function in Eq (13) of [4]:*

$$H(\boldsymbol{W}, \hat{\boldsymbol{M}}) = aL(\boldsymbol{W}) + \frac{1}{1-b}\mathcal{K}((1-b)\hat{\boldsymbol{M}}).$$

$$\frac{\mathrm{d}}{\mathrm{d}t}H(\boldsymbol{W}_t, \boldsymbol{M}_t) = a\langle\boldsymbol{G}_t, \boldsymbol{P}_t\nabla\mathcal{K}((1-b)\hat{\boldsymbol{M}}_t - b\hat{\boldsymbol{G}}_t)\rangle - a\langle\nabla\mathcal{K}((1-b)\hat{\boldsymbol{M}}_t), \hat{\boldsymbol{G}}_t + \hat{\boldsymbol{M}}_t\rangle$$

$$= a\langle\hat{\boldsymbol{G}}_t, \nabla\mathcal{K}((1-b)\hat{\boldsymbol{M}}_t - b\hat{\boldsymbol{G}}_t) - \nabla\mathcal{K}((1-b)\hat{\boldsymbol{M}}_t)\rangle - a\langle\nabla\mathcal{K}((1-b)\hat{\boldsymbol{M}}_t), \hat{\boldsymbol{M}}_t\rangle$$

$$= -\frac{a}{b}[(1-b)\hat{\boldsymbol{M}}_t; \; -b\hat{\boldsymbol{G}}_t]_{\nabla\mathcal{K}} - \frac{a}{(1-b)}[\boldsymbol{0}; \; (1-b)\hat{\boldsymbol{M}}_t]_{\nabla\mathcal{K}} \le 0$$

*where we defined* $[\boldsymbol{X}; \boldsymbol{Y}]_{\nabla\mathcal{K}} = \langle\boldsymbol{Y}, \; \nabla\mathcal{K}(\boldsymbol{X}+\boldsymbol{Y}) - \nabla\mathcal{K}(\boldsymbol{X})\rangle$ *and used the fact that* $[\boldsymbol{X}; \boldsymbol{Y}]_{\nabla\mathcal{K}} \ge 0$ *by the convexity of* $\mathcal{K}$; *we used* $\langle\boldsymbol{G}_t, \boldsymbol{P}_t\boldsymbol{X}_t\rangle = \langle\boldsymbol{P}_t^\top\boldsymbol{G}_t, \boldsymbol{X}_t\rangle = \langle\hat{\boldsymbol{G}}_t, \boldsymbol{X}_t\rangle$.

In all examples above, although the form of Hamiltonian $H(\boldsymbol{W}, \hat{\boldsymbol{S}})$ is independent of the update rule of $\boldsymbol{P}_t$, the decreasing rate $\frac{\mathrm{d}}{\mathrm{d}t}H(\boldsymbol{W}_t, \hat{\boldsymbol{S}}_t)$ depends on $\boldsymbol{P}_t$ in a complicate way through $\hat{\boldsymbol{M}}_t, \hat{\boldsymbol{V}}_t,$ $\hat{\boldsymbol{G}}_t$. An interesting direction for future investigation is to find optimal rules of $\boldsymbol{P}_t$ to maximize the decreasing rate as an optimal control problem.

# B Experiments

## B.1 Hyperparameters

We sweep learning rate from [0.01, 0.005, 0.001]. For GaLore as well as Adam8bit, we follow the recommended hyperparameters as much as possible. For instance, GaLore update gap is set to recommended default, 200. Warmup is set to $10\%$ of the total training steps. Batch size is set to 512 and gradient clipping is set to 1.0.

## B.2 Rank Sweep

In the following table, is a sweep on different ranks and their final perplexity of LLaMA 60M (SS = 1024) on C4. All other hyperparameters are fixed and using recommended default. Notice that as the rank increases, both Dynamic Subspace Descent and GaLore improve.

| Rank | Perplexity (Ours) | Perplexity (GaLore) |
|------|-------------------|---------------------|
| 32   | 85.90             | 86.16               |
| 128  | 49.01             | 48.05               |
| 512  | 37.41             | 36.93               |
| Full | 37.18             | 36.51               |

Table 5: On LLaMA 60M SS 1024, we sweep across different ranks, the trend is clear and intuitive that higher rank is preferred when it's feasible.

## B.3 Optimizer Sweep

| Method | Perplexity |
|--------|------------|
| Lion | 52.65 |
| Adafactor | 33.45 |
| Adam8bit | 29.77 |
| SGD | 3469.14 |
| Galore LION | 46.90 |
| Galore Adafactor | 34.32 |
| Galore AdamW8bit | 48.05 |
| Ours LION + LION | 57.97 |
| Ours Adafactor + Adafactor | 47.61 |
| Ours AdamW8bit + AdamW8bit | 49.01 |
| Ours LION + Adam8bit | 44.76 |
| Ours Adafactor + AdamW8bit | 34.15 |
| Ours AdamW8bit + SGD | 53.53 |

Table 6: In this experiment, we train LLaMA 60M on C4 with sequence length of 1024. We combine different base optimizers to update both $P_t$ and $W_t$, denote as "Ours weight optimizer + $P_t$ optimizer".

