# OpenReview forum: "Memory-Efficient LLM Training with Online Subspace Descent"
_NeurIPS.cc/2024/Conference — NeurIPS 2024 poster_

### Official Review · Reviewer_FtMv · 2024-06-29

**Soundness:** 2
**Presentation:** 3
**Contribution:** 2
**Rating:** 5
**Confidence:** 3

**Summary:**

The paper considers memory-efficient optimization for large language model. In particular, the focus is on optimizers leveraging low-rank projection. The paper first derives asymptotic convergence for such methods with arbitrary choice of projection matrix. Then the paper identifies the inefficiency of using SVD to construct the projection matrix and proposes online subspace descent method by updating the projection matrix with online PCA. Experiments show such method improves the efficiency of training LLM while achieving comparable performance compared to low-rank baseline with SVD projection.

**Strengths:**

1. The paper derives the first convergence result for optimizers with low-rank projection.

2. The paper addresses the inefficiency of using SVD to construct the projection matrix by online subspace descent.

3. Experiments show the benefits of the online subspace descent over the SVD baseline.

**Weaknesses:**

1. The main downside of the proposed method is that it could potentially increase the memory overhead. It seems to me that the idea of online updating $P_t$ trades off computation with memory. Although it reduces the computational cost of not computing SVD, it nevertheless increases the memory especially using Adam type of optimizer for $P_t$. In Line 216, the paper states the memory overhead is 9.01GB versus 8.64GB. However, it is not clear what model this numbers refer to. I suspect with the increase in rank and model size, the memory overhead could become larger.

2. The use of Hamiltonian descent and Lyapunov analysis for showing convergence is not properly motivated. Because the main result (Theorem 4.5) only shows asymptotic convergence to stationary point of $L(W)$, it is not convinced why to derive such results under continuous time limit. Can such result also be derived in the discrete case and can non-asymptotic result be derived?

3. Figure 2 is not a fair comparison because in practice, SVD is not used every update.

**Questions:**

1. In Line 233, what does it mean to be high-frequency versus low-frequency tokens? And is there any verification or justification that high-frequency tokens are learned with low-rank training while learning low-frequency tokens requires higher ranks?

2. In Line 252, can you comment further why AdamW is more suitable for online PCA?

**Limitations:**

The paper has discussed limitations.

---

> ### Author Rebuttal · Authors · 2024-08-07
>
> ## Re: “Memory and efficiency”
> Updating $P_t$ with AdamW will inadvertently increase the memory consumption. However, in practice, we observe that the increase in peak memory is minuscule compared to the overall size of the model. Meanwhile, we can enjoy the nice properties of online subspace descent for its flexibility and system efficiency when implemented in parallel (as we discussed with reviewer NLbB and Rbsr). In addition, our subsequent experiment shows that SGD can perform similarly to AdamW when updating P, in which case, there is no additional memory cost over GaLore, since no momentum needs to be saved.
>
> Here is a comparison between GaLore and our method on 7B llamas trained for 10K steps on C4 (updating $P_t$ with SGD)
> | Method | Perplexity | Wall Clock Time (hours) |
> |--------|------------|-------------------------|
> | Galore | 51.21      | 9.7439                  |
> | Ours   | 43.72      | 7.1428                  |
>
> The hardware setups are completely identical, both using a single node of H800s with 8 cards. It's clear that our method is not only faster in wall clock time, but also reaches lower perplexity when seeing same number of tokens.
>
> ## Re: "Discrete case"
> The derivation of discrete case is highly similar to the continuous example we've shown in the paper.
>
> **Example 4.1.** Momentum + Online Subspace Descent is
>
> $$
> \frac{d}{dt} W_t = -P_t \hat{M}_t, \quad \hat{G}_t = P_t^\top \nabla L(W_t), \quad \frac{d}{dt} \hat{M}_t = a(\hat{G}_t - \hat{M}_t),
> $$
>
> with Lyapunov function $H(W, \hat{M}) = L(W) + \frac{\|\hat{M}\|^2}{2a}$, for which
>
> $$
> \frac{d}{dt} H(W_t, \hat{M}_t) = - \nabla L(W_t)^\top P_t \hat{M}_t + \hat{M}_t^\top (P_t^\top \nabla L(W_t) - \hat{M}_t) = -\|\hat{M}_t\|_2^2 \leq 0.
> $$
>
> Followup this example as a demo, we have the following discrete derivation:
>
> $$W_{t+1} = W_t - \epsilon P_t \hat{M}_t$$
>
> $$\hat{M}_{t+1} = \hat{M}_t + a(\hat{G}_t - \hat{M}_t)$$
> $$ \hat{G}_t = P_t^\top \nabla L(W_t)$$
>
> where $\hat{M}_{t+1} = \hat{M}_t + a(\hat{G}_t - \hat{M}_t) = (1-\alpha \epsilon)\hat{M}_t + \alpha \epsilon \hat{G}_t = \beta \hat{M}_t + (1-\beta)\hat{G}_t$.
> Assuming $L(\cdot)$ is $L$-smooth, we have
>
> $$
> H(W_{t+1}, \hat{M}_{t+1}) - H(W_t, \hat{M}_t) \leq -\epsilon \|\hat{M}_t\|_2^2 + \alpha \epsilon^2 \|\hat{G}_t - \hat{M}_t\|_2^2 + \frac{L}{2}\epsilon^2\|P_t\hat{M}_t\|_2^2.
> $$
>
> Hence, a telescoping sum yields
>
> $$
> \frac{1}{T}\sum_{t=0}^{T-1} \|\hat{M}_t\|_2^2 \leq \frac{H(W_0, \hat{M}_0) - H(W_T, \hat{M}_T)}{\epsilon T} + \epsilon B_t,
> $$
>
> where $B_t = \frac{1}{T} \sum_{t=0}^{T-1} \alpha \|\hat{G}_t - \hat{M}_t\|_2^2 + \frac{L}{2}\|\hat{M}_t\|_2^2$.
>
> ## Re: “high-frequency versus low-frequency tokens”
> This is our attempted explanation for the phenomenon we observe in our ablation study of different ranks. Loss of higher rank runs tend to drop faster and converge to a lower loss after 10K steps. We suspect that it’s due to the unique property of language modeling tasks, where intuitively rarer tokens (tokens that are not common) require high gradient rank to be learnt. However, we would like to delay the justification of this hypothesis to future study, since it’s not the main focus of our paper.
>
> ## Re: "AdamW is more suitable for online PCA"
> This is what we find empirically, when trying different optimizers such as LION and Ada Factor. However, it’s not unfathomable that SGD in principle can perform similarly, which we show above. It’s just for the default learning rates and schedules, adamW seems to work more robustly and reliably and it’s the least sensitive one.

---

> > ### Comment · Reviewer_FtMv · 2024-08-11
> >
> > Thank you for the responses. For the discrete-time convergence analysis, first, the convergence is measured in terms of gradient momentum, which is non-standard. Second, there seems to be an extra term $\epsilon B_t$, which depends on the entire past history as well as the gradient momentum, which is also on the left hand side of the final inequality. Can the authors clarify?

---

> ### Author Response · Authors · 2024-08-11
>
> The derivation presented above provides a foundational outline for our convergence analysis. Building on this, we can conduct a more detailed analysis following the established classical (or traditional) approach.
>
> Taking a step further from the bound above, we have
>
> $(1 - \frac{L \epsilon}{2}) \frac{1}{T} \sum_{t=0}^{T-1} ||\hat{M}_t||_2^2 \leq \frac{H(W_0, \hat{M}_0) - H(W_T, \hat{M}_T)}{\epsilon T}$
>
> $+\epsilon \frac{1}{T} \sum_{t=0}^{T-1} \alpha ||\hat{G}_t - \hat{M}_t||_2^2$
>
> One can use the established standard methods to bound the last term $||\hat{G}_t - \hat{M}_t||_2^2$ (for example at page 11-12 in the supplementary from [Sun et al. 2023]).
>
> ### References
>
> Tao Sun, Qingsong Wang, Dongsheng Li, and Bao Wang. Momentum ensures convergence of signsgd under weaker assumptions. In *International Conference on Machine Learning*, pages 33077–33099. PMLR, 2023.

---

### Official Review · Reviewer_RBsr · 2024-07-12

**Soundness:** 3
**Presentation:** 3
**Contribution:** 3
**Rating:** 5
**Confidence:** 3

**Summary:**

The authors propose a variation of GaLore that doesn't require a fixed SVD for achieving a memory efficient LLM training. Instead, the authors project the gradients dynamically into a small sub-space using online PCA which depends on the evolving trace of the gradient landscape.

The authors tested their method against GaLore using perplexity as a metric while training on the C4 dataset.

**Strengths:**

- the paper is well written and easy to follow
- having a dynamically changing projection matrix based on the gradient landscape makes total sense
- the theoretical explanation shows an interesting perspective on why this proposed method of using online pca for matrix projection could work
- the results is promising in C4 as the memory seems more or less similar in efficiency as GaLore with similar perplexity.

**Weaknesses:**

- the perplexity results is not enough to show that the method works well on standard NLP tasks like GLUE which consists of the datasets MRPC, COLA, STS-B, RTE, SST2, MNLI, QNLI, QQP to see if the actual downstream task performance is still retained with the proposed projection method.
- while the theory is interesting, it seems to include many strong assumptions that do not reflect real-world datasets
- the memory savings and the perplexity is very similar to GaLore and it is not clear that it is significant. The authors need to do multiple runs and compute the standard deviation to make the results are not due to noise.

**Questions:**

please answer how you could address the weaknesses above

**Limitations:**

Yes

---

> ### Author Rebuttal · Authors · 2024-08-07
>
> ## Re: "Downstream tasks"
> We did standardized GLUE evaluation for the above two 7B checkpoints with eval-harness.
>
> | Method | MRPC  | RTE   | SST2  | MNLI  | QNLI  | QQP   | AVG    |
> |--------|-------|-------|-------|-------|-------|-------|--------|
> | GaLore | 0.6838| **0.5018**| 0.5183| 0.3506| 0.4946| 0.3682| 0.4862 |
> | **Ours**   | **0.6982**| 0.4901| **0.5233**| **0.3654**| **0.5142**| **0.3795**| **0.4951**|
>
> Both Cola and STSB require further fine-tuning, which brings in more confounding factors (even Llama-2 7B trained by Meta doesn't get meaningful results), so we exclude them from the comparison.
>
> ## Re: “Many strong assumptions that do not reflect real world datasets”
> We respectfully argue that our assumptions (Assumption 4.4) are in fact, very much minimum.
> 1) The model needs to be continuously differentiable. (pretty much all modern neural nets)
> 2) The optimization stops when the projected update hits 0. (this is rather a fact than a assumption)
> 3) It says we should pick $P$ such that $P^TG$, the projected gradient is not always 0. Otherwise, it just doesn’t update the model weights at all and it defeats the purpose.
>
> Every theoretical work has to make some assumptions. We want to assure our reviewers that it will be hard to find papers with same level of theoretical rigorousness that is making fewer assumptions than we do: we did not assume convex losses; we did not assume specific forms of the model (linear model or MLPs); we did not even assume specific optimizers (Adam, momentum, lion and etc)...
>
> ## Re: “The memory savings and the perplexity is very similar to GaLore and it is not clear that it is significant”
> We agree that this does have very similar memory saving as GaLore. However, we argue that our contributions lie majorly in our analysis that introduces a new family of optimizers that are guaranteed to converge. Among them, online subspace descent not only possesses the same memory saving properties, but also can be implemented more efficiently in a parallel manner, whereas Galore doesn’t share the same flexibility.

---

> > ### Comment · Reviewer_RBsr · 2024-08-14
> >
> > Thank you for the rebuttal and for addressing most of my concerns about the assumptions and memory saving. I have increased my score by 1.

---

### Official Review · Reviewer_6zai · 2024-07-13

**Soundness:** 3
**Presentation:** 3
**Contribution:** 3
**Rating:** 6
**Confidence:** 2

**Summary:**

Utilizing low-rank structure has become a popular way for memory-efficient LLM training. The authors are the first to provide convergence guarantees for general low-rank updating rules. Furthermore, based on their theoretical result, they propose a family of optimizers called online subspace descent. The empirical result shows that their performance is better than existing low-rank training methods.

**Strengths:**

This work provides a general theoretical guarantee for a range of low-rank updating rules, which can guide and help future research in this direction.

**Weaknesses:**

While the authors provide mathematical proofs, it could be more beneficial to the readers if they can provide more high-level intuitions why these methods could converge. As stated by the authors, the convergence result is surprising under the complications.

**Questions:**

Could you provide more high-level intuitions on why the methods could converge under the complications?

**Limitations:**

Yes, the authors discuss the limitations.

---

> ### Author Rebuttal · Authors · 2024-08-07
>
> ## Re: “Intuitions”
> The rough intuition is that $P_t$ serves as a kind of preconditioning matrix in the Hamiltonian systems. But to arrive at the precise mathematical conclusion, we find that the best and quickest way to understand it is through the derivation in Eq (8), together with physical understandings of Hamiltonian systems (that, e.g., the hamiltonian consists of potential energy and kinetic energy, and the kinetic energy term is different for different optimizers)
>
> More specifically, as long as the projected subspace defined by $P_t$ keeps changing and not always be orthogonal to the gradient as we stated in Assumption (4.4), it will always make some progress towards the fixed point. Given enough time, it will eventually reach there. Hence the consequence of convergence does not depend on $P_t$.

---

> > ### Comment · Reviewer_6zai · 2024-08-11
> >
> > Thank you for the response to my question!

---

### Official Review · Reviewer_NLbB · 2024-07-14

**Soundness:** 3
**Presentation:** 3
**Contribution:** 3
**Rating:** 7
**Confidence:** 4

**Summary:**

The paper presents Online Subspace Descent, a memory-efficient modification applicable to a wide class of gradient-descent based algorithms where low-rank projections of gradients can be employed to reduce the memory overhead. Contrary to recent techniques such as GaLore, which require infrequent but costly updates of the projection matrix, the proposed method updates the projection matrix in an on-line manner, and proves the convergence of the resulting algorithm.

**Strengths:**

Memory-efficient optimizers are crucial to enable LLM training/fine-tuning outside of huge datacenters; as such, this is a potential high-significance paper. While I think the idea of gradually updating the projection matrix is pretty much "the obvious thing to do", I really like that the authors didn't just propose "Adam[Add-some-random-letter-here]", but instead took a step back and showed that the same principle applies to a wide class of optimization algorithms.

**Weaknesses:**

* There are some grammatical errors and typos, I think (not a native speaker), but not to the degree that they hamper the understanding of the text (e.g., I think it should be: update rules of _the_ projection matrix; in the subspaces that dynamically _change_, How Should P be _updated_, is not a problem for optimizers that _yield_)

l.1ß6: `Note that the update of P t can be done in parallel with that of 106 (W t, ˆ St), and incurs no slow down once it is fast enough to not cause a speed bottleneck` Generally, Adam updates are memory-bound operations, so I'm not sure if you can schedule a second operation concurrently without slowing down the update. Also, the same could be said for GaLore, you run you SVD in parallel to the main network; if it takes multiple steps, you could just wait till it is done, and update only afterwards, so GaLore can be made as parallel as the algorithm that is proposed here, I think.

l.119: `In this work, we propose to update P t in a continuous online fashion that incorporates the most 119 recent gradient information in a timely fashion, without calling linear algebra routines.`
Matrix multiplications are linear algebra routines, so I think most P updates will involve some linear algebra.

I think the description around l. 119-128 could do with some rewriting. The sentence `we propose to update P t at time t by minimizing the following PCA objective` is misleading, because P_t is not actually updated like that; instead P_t is one step along some optimizer that decreases this objective. Second, `Instead of minimizing [...] exactly, to retain computational efficiency, we propose` seems to be misleading, too; in an earlier paragraph, it was GaLore was criticized for using only a single gradient information for its update. This sentence suggests that if you had enough compute, you would do the same here, only due to computational constraints do you end up with something that integrates multiple gradients, which I think goes against the message you want to make in the paper.

l. 170:  `There is no requirement on Γ here, besides that the derivative in (8) should exist` Γ does not appear in (8), so it isn't clear which derivative should exist.

l. 223: `The typical Pytorch implementation of SVD can be up to 142 times slower than running a single step online PCA on representative weight tensors from LLaMA architectures.` Given that GaLore runs SVD only every 300 steps, this would make GaLore comutationally more efficient than the proposed method.

**Questions:**

It has been observed that scaling up LLMs is notrivial also from a numerical stability point of view, e.g., in quantization, starting with 7B models, outliers become much more of a problem than they are for smaller models. As such, it would be great to see this method applied to larger models, at least to 7B, ideally even larger. If compute is a problem, maybe scaling the amount of training steps less than would be optimal could help, to at least have some proof-of-concept for the method at larger model scales.


l.145: `this is a mild condition that can be satisfied e.g., when P_t is updated by (online) PCA on G_t` Is this really true, in the generality that is suggested here? If so, I'd like to see a formal argument. The paper only fixes the objective eq. (6), but leaves the choice of optimizes to the user, and my intuition says it should be possible to construct an example where minimizing (6) with a momentum-based optimizer could lead to $<G_t, P_t>$ being zero without $G_t$ being zero.

l 158, I'm not sure what "once" is supposed to mean here. Should it just be "if"?

Regarding the convergence guarantee for arbitrary $P_t$, I think it would be illuminating to consider what happens in case of an adversarial choice; for example, for SGD with momentum, choose $P_t = min <G_t, P_t^T P M_t>$, i.e., choose P so that it projects the momentum buffer _against_ the direction of the current gradient. Why does this choice of P not break convergence?

l. 235
> In conclusion, given sufficient time and resources, higher ranks yield better performance for Online Subspace Descent. It is recommended to select the highest rank until the perplexity reduction saturates.

Can this reliably be selected a priori, i.e., does picking a rank optional for running optimization for, e.g., 10k steps, also result in optimal rank for training until "convergence"

**Limitations:**

Evaluations for production-size LLMs are missing.

---

> ### Author Rebuttal · Authors · 2024-08-07
>
> ## l.1ß6 “Generally, Adam updates are memory-bound operations”
> - This is precisely the bottleneck that this family of online subspace descent optimizers aims to tackle. By projecting into subspace, the memory footprint of Adam will be greatly reduced, allowing us to schedule another operation concurrently.
> - In addition, since we are running tensor-wise operation, the incremental memory consumption over running adamW in subspace is simply backward of P for a single tensor. This is reflected in our experiment for Llama 1B, where the peak memory increase compared with Galore AdamW is only 0.4 GB, that is counting fragmentations, redundant intermediate saving in practice.
>
> ## l.1ß6 “the same could be said for GaLore, you run you SVD in parallel to the main network”
> - It can definitely run SVD in parallel synchronously. But it won’t benefit from doing that. Instead, it will slow down that specific training step SVD is paralleled with if SVD on the latest gradient is computed. SVD uses Jacobi iterations until the principal components converge, in some cases it fails and has to restart. Overall on weight matrix sizes we care about, we observe that  it’s 20x - 150x slower than running a single backward step + update which makes it impossible to be hidden by the adamW operations. Waiting for SVD to be done slows down training, whereas we essentially propose to distribute and amortize that workload into every step of training instead of doing it all at once and wait at a single step. In our latest experiment of the 7B model, we also show that our method is 1.37x faster than baseline GaLore in wall clock time.
> - In another scenario, it’s also possible to run SVD completely asynchronously and use $P_t$ computed from some arbitrarily old gradients to perform projections and updates. However, this becomes a very different class of algorithms from baseline GaLore. It overly complicates implementation in practice, since it needs to manage schedules for every tensor in the network. SVD finish time can also vary and it is unpredictable. Hence, it's possible to have part of the network using $P_t$ calculated from a different timestep, if it's running in a non-blocking fashion. The implication of this is unknown and it's a very interesting direction of exploration. Yet, given the limited span of this work, we would like to defer this discussion to future works.
>
> ## l.119 "Matrix multiplications are linear algebra routines"
> We meant to refer to torch.linalg.decomposition routines. We will make it more specific in revision.
>
> ## Re: "119-128 could do with some rewriting"
> - We agree that we can rewrite it to highlight the goal of doing online PCA, not just for saving computational cost.
>
> ## l. 170 "There is no requirement on Γ here, besides that the derivative in (8) should exist Γ does not appear in (8), so it isn't clear which derivative should exist."
> - The derivatives in (8) refers to the partial derivatives of $H_t$ that appear in Eq (8). We will make it specific in the revision.
>
> ## l. 223 "Given that GaLore runs SVD only every 300 steps, this would make GaLore computationally more efficient than the proposed method."
> Please refer to the above explanation on system efficiency/parallel implementations. Even if running Galore every 300 steps, it’s still an unmitigable slowdown where no parallelization tricks can help. And as tensor shape grows, as figure 2 shows, the gap of latency becomes higher. Meanwhile the cost of online subspace descent can be hidden with a smart parallelization trick.
> Our latest experiment of 7B model shows that our method is 1.37x faster than baseline GaLore in terms of wall clock time with the exactly the same hardware setup.
>
> ## Re: "Larger model"
> We pretrained from scratch 7B llamas on C4 for 10K steps. Perplexity is the lower the better. The results are consistent with our claim in production model scale.
> | Method | Perplexity | Wall Clock Time (hours) |
> |--------|------------|-------------------------|
> | Galore | 51.21      | 9.7439                  |
> | **Ours**   | **43.72**      | **7.1428**                  |
>
> ## l.145 " Adversarial projections"
> The loss in $L_{G_t}(P)$ attains global optima only if P spans the linear space of the top k eigenvectors of $G_t$ in Loss (the other eigenvectors are saddle points). Hence, if an optimizer minimizes $L_{G_t}(P)$ correctly, it should find the top k eigenvectors of G, which is zero iff G ==0. Regardless of the choice $P_t$, the system yields a given Hamiltonian function and converges to some invariant set. But the invariant set coincides with the local optima of the objective function f only when Assumption 4.4 satisfies. The choice you provide may not satisfy Assumption 4.4.
>
> More concretely, solution of $P_t=min<G_t,P_t^TPM_t>$ gives the opposite direction of $G_t$. Considering the negative direction won’t harm it, because it’s still the same subspace even the gradient has an opposite sign and while it’s projected back, it’s still “gradient descent”, the two negative signs simply cancel out. The real adversarial scenario is $P_t=min|<G_t,P_t^TPM_t>|$. It is projecting into complementary orthogonal subspace, aka, $<G_t,P_t^TPM_t> = 0$. This is against ii) in our Assumption 4.4, which states we can not pick $P$ such that $P^TG$, the projected gradient is always 0. Otherwise, it just doesn’t update the model weights at all and it defeats the purpose.
>
> ## l 158 "I'm not sure what "once" is supposed to mean here. Should it just be "if"?"
> Yes, it should be “if”. We will correct that in revision.
>
> ## l. 235 "choices of rank"
> Just in terms of speed of convergence, in our ablation study, within 10K steps, the higher rank it’s the faster it goes. Eventually, if it’s run long enough, our theory indicates that it will always converge. However, in practice, it might be difficult to observe that due to constraint in resources. So as a safe recommendation, if it can be afforded, choose a higher rank.

---

> > ### Comment · Reviewer_NLbB · 2024-08-13
> >
> > Thank you for the response, which adequately addresses the concerns raised in my review. With the additional clarifications as described, and an additional experiment in the regime where memory-efficient optimizers start to become very important and useful, I think this is a good paper that, were I to see it for the first time at the conference, I probably would point out to my colleagues. I will raise my score to an "Accept" rating.

---

### Author Rebuttal · Authors · 2024-08-07

We really appreciate our reviewers for their constructive reviews and suggestions. Here we summarize and highlight our response to a few main points of concerns. Hope that will help address the majority of those questions.

## Re: “System efficiency compared to Galore” (NLbB, FtMV, RBsr)
In terms of speed, online subspace descent’s additional cost can be easily hidden by simple parallelization, whereas GaLore can not. As model size increases, the gap in latency between our method and GaLore becomes larger. Additional experiment shows on 7B model on same hardware setup, our method can be up to **1.3x** faster than GaLore.

In terms of memory, we have flexibility in choosing different optimizers for updates of $P_t$. When choosing AdamW, inadvertently, it increases memory consumption slightly over GaLore. We chose to start with AdamW due to its robustness to hyperparameters. However, additional experiment on the 7B model shows that SGD also shares similar properties and can indeed be used for $P_t$ updates, incurring no extra memory cost over GaLore.

## Re: “Larger scale experiment” (NLbB)
We pretrained from scratch 7B llamas on C4 for 10K steps. Perplexity the lower the better. The results are still consistent with our claim in production model scale. $P_t$ is updated by SGD.
| Method | Perplexity | Wall Clock Time (hours) |
|--------|------------|-------------------------|
| Galore | 51.21      | 9.7439                  |
| **Ours**   | **43.72**      | **7.1428**                  |

## Re: “Downstream tasks” (RBsr)
We did standardized glue evaluation for the above two 7B checkpoints with eval-harness.
| Method | MRPC  | RTE   | SST2  | MNLI  | QNLI  | QQP   | AVG    |
|--------|-------|-------|-------|-------|-------|-------|--------|
| GaLore | 0.6838| **0.5018**| 0.5183| 0.3506| 0.4946| 0.3682| 0.4862 |
| **Ours**   | **0.6982**| 0.4901| **0.5233**| **0.3654**| **0.5142**| **0.3795**| **0.4951**|

Both Cola and STSB require further fine-tuning (even Meta Llama-2 7B gets unmeaningful results), so we exclude them from the comparison.

## Re: “Motivation for the Lyapunov Analysis” (FtMV, 6zai)
The rough intuition is that Pt serves as a kind of preconditioning matrix in the Hamiltonian systems. But to arrive the precise mathematical conclusion, we find that the best and quickest way to understand it is through the derivation in Eq (8), together with physical understandings of Hamiltonian systems (that, e.g., the hamiltonian consists of potential energy and kinetic energy, and the kinetic energy term is different for different optimizers)

---

### Decision · Program_Chairs · 2024-09-25

**Decision:**

Accept (poster)

**Comment:**

After the author response, all reviewers ultimately recommended acceptance of the paper.